# Caloric Vestibular Stimulation Reduces the Directional Bias in Representational Neglect

**DOI:** 10.3390/brainsci10060323

**Published:** 2020-05-26

**Authors:** Julie Holé, Karen T. Reilly, Stuart Nash, Gilles Rode

**Affiliations:** 1Service de médecine physique et réadaptation, Hôpital Henry-Gabrielle, Hospices Civils de Lyon, 69610 Pierre-Bénite, France; stuart.nash@chu-lyon.fr; 2ImpAct team, Lyon Neuroscience Research Center, INSERM U1028, CRNS-UMR5292, 69675 Bron, France; karen.reilly@inserm.fr; 3Lyon 1 University, 69008 Lyon, France

**Keywords:** representation neglect, vestibular stimulation, mental representation, hemispatial neglect, unilateral spatial neglect, hemineglect

## Abstract

Caloric vestibular stimulation (CVS) can temporarily reduce visuospatial neglect and related symptoms. The present study examined the effect of CVS on representational neglect during free exploration of the map of France. We asked patients to name cities they could mentally “see” on the map of France, without giving them any directional instructions related to the left or right sides of the map. In right brain damaged patients with left visuospatial neglect, the mental representation of the map was asymmetrical (favoring the right side). After stimulation, neglect patients named more towns on the left side of the map, leading to a significant reduction in map representation asymmetry. Our findings are consistent with previous studies on visuospatial neglect and are in favor of a central effect of vestibular stimulation on mechanisms involved in space representation.

## 1. Introduction

Unilateral spatial neglect is a cognitive disorder defined as a failure to report, to respond to or to orient to contralesional stimuli [1]. This deficit can affect different parts of the physical contralesional space [2]. Since the pioneering study of Bisiach and Luzzatti, it is well known that unilateral spatial neglect can also affect representational space [3]. These authors described two right brain damaged patients with difficulties describing, imagining and mentally exploring the Piazza del Duomo in Milan from a set point of view. The patients reported a larger number of landmarks on the right side of the imagined square. This bias was present both when they imagined themselves standing with their back to the cathedral and when they were facing the cathedral from the opposite end of the square. This inner bias was interpreted as a representational deficit, seen as a failure to generate or maintain a normal representation of the left side of the mental image [4,5]. 

Several propositions have been put forward in an attempt to account for this bias (see review [6]), including a deficit in generating mental representations (the possible consequence of a visuospatial working memory deficit [7]), or an ipsilesional bias in spatial attention orientation (which results in difficulty exploring the contralesional side of imaginary representations [8,9]). The spatial attention orientation proposition is supported by the fact that when patients with left-sided neglect after right brain damage are asked to turn their head towards the neglected side they report more items on the left side of the imagined scene than when instructed to turn their head to the right [10]. Finally, this bias could arise from a deficit in processing representations from a body-centered vantage point, that is, using an egocentric frame of reference [11,12]. Indeed, there is substantial data indicating that unilateral spatial neglect may be related to a lateral deviation of an egocentric frame of reference [13,14,15]. 

Although the mechanism underlying neglect is still a matter of debate, it can be alleviated using sensory or sensory-motor manipulations like caloric vestibular stimulation (CVS) (see review [16]). The effects of CVS on representational neglect were first tested by Geminiani and Bottini in five right brain damaged patients with left neglect [17]. They asked them to imagine and mentally explore the Piazza del Duomo in Milan taking two opposite vantage points. Patients’ performance improved after CVS regardless of their vantage point, with a return to pre-stimulation levels when re-tested the following day. Similar findings, with better performance after CVS, have been reported in right brain damaged patients with left neglect asked to mentally evoke the map of France and to name all the towns they could “see” on the right, then on the left, side of the map and vice versa [18]. Despite differences in the two tasks, both the Italian and French tasks revealed a similar representational deficit and transient alleviation of the deficit following CVS. Indeed, while the Piazza del Duomo corresponds to a familiar three-dimensional space, the map of France task involves a familiar two-dimensional geographical space. Furthermore, since the Milanese patients’ representation of the Piazza del Duomo is likely based on personal recollection experienced in a precise spatial and temporal context, the Geminiani and Bottini study called upon verbal episodic memory, whereas semantic knowledge was used in the French study as the French patients had all learned the map of France in primary school. The similarities in the results, despite these differences, suggest that both tasks reveal a common underlying disorder selectively alleviated by CVS, thereby supporting the idea that CVS acts via a higher-level central mechanism.

The aim of this study was to further our understanding of representational neglect. To do this we tested eight right-brain damaged patients with left neglect under conditions in which they were completely free (i.e., received no directional instructions) to imagine, mentally explore and name as many cities as they could locate on the map of France before and after CVS. In the previous study using the map of France, patients were given a fixed amount of time (1 min) to explore the right side of the map, after which they then had 1 min to explore the left side (the reverse order was also tested). The degree of the bias and the effect of CVS were not the same as a function of the direction of exploration, suggesting that directing patients’ attentional orientation interferes with the representational deficit and the effect of CVS on this deficit. In this study we wanted to investigate the effect of CVS in the absence of directional instructions in order to explore the full potential of CVS on representational neglect.

## 2. Materials and Methods

### 2.1. Participants

The study included eight right brain damaged patients who exhibited left unilateral neglect (RBD+) and three aged-matched control groups each composed of eight participants. Control participants were (1) patients with right brain damage without neglect (RBD−), (2) patients with left brain damage without neglect (LBD−) and 3) healthy subjects (CONT). The age and gender of each of the four groups are presented in Table 1. All participants were right-handed and provided informed consent. The study was approved by the ethical committee of the Hospices Civils de Lyon and was conducted in accordance with the Declaration of Helsinki and carried out according to the principles of medical research.

Patients in all three groups were admitted to the department of neurological rehabilitation (Hôpital Henry Gabrielle, Hospices Civils de Lyon) for rehabilitation after a stroke. The inclusion criteria for all patients were: age between 20 and 80 years, a single stroke confirmed by a tomodensitometry examination, a delay post-onset of less than six months following the stroke and the ability to perform the mental imagery map of France task. Exclusion criteria were: existence of multiple brain lesions, disorientation in time and space (determined by questioning patients during a clinical examination), psychiatric disorders and a co-existing non-stable pathology. Patients in the RBD+ group had to show signs of unilateral spatial neglect while patients were excluded from the RBD− and LBD− control groups if they had initially shown signs of neglect or if they showed any signs at the time of testing. 

Unilateral neglect was objectified by a thorough neuropsychological assessment, which included:-peripersonal neglect using the Albert line cancellation test with a threshold of 0 omissions [19] and the Schenkenberg line bisection test with a mean deviation of more than 6 mm or omission of 2 or more lines indicating neglect [20]-personal neglect using the Bisiach score [21]-object centered neglect assessed by coping and drawing from memory-oculocephalic deviation toward the right side using Geren scale [22]-anosognosia for motor deficit using clinical questionning and the Bisiach score [23]. Scores are assigned in the following way: 0: disorder spontaneously reported; 1: disorder reported following a specific question; 2: disorder acknowledged after having been demonstrated through routine techniques of neurological examination; 3: disorder not acknowledged.

Patients were considered to have neglect if neglect was detected on at least one of the following tests: Albert line cancellation test, Schenkenberg line bisection test or Bisiach score for personal neglect. Left hemibody motor deficit was assessed by a clinical examination and a range of other deficits were investigated, including constructional apraxia (copy of a cube and drawing from memory), dyslexia (reading a text), hyperschematia (previous drawing from memory) and motor perseveration on a cancellation test and drawing from memory. The results of these tests for the RBD+ group are reported in Table 2. All patients in the RBD+ group showed signs of peripersonal neglect; 4 patients had personal neglect and 3 patients had an anosognosia for motor and visual deficits. 

None of the patients in the RBD− and LBD− control groups showed an oculo-cephalic deviation, nor signs of unilateral spatial neglect assessed by the neuropsychological examination performed at the time of the experiment. Lesions were either ischemic (*n* = 11) or hemorrhagic (*n* = 5), and all were in the territory of the middle cerebral artery. Four patients in each group displayed a hemianopia. 

### 2.2. Procedure

All participants were asked to mentally evoke the map of France as if it were in front of them and to name all the towns they could “see” within two minutes while keeping their eyes open. Three scores were obtained according to the geographical position of the named towns: (1) number of towns named within the central axis of the map (towns located inside a 75 km strip centered on the Lille-Perpignan axis), (2) number of towns named on the left of this strip, (3) number of towns named on the right of this strip. Patients and control subjects performed this task once, while RBD+ patients performed it before and immediately after CVS. 

The three RBD+ patients with anosognosia (see Table 2) also underwent a clinical examination for anosognosia immediately following their post-CVS map of France test. Due to the short lasting nature of the effect of CVS (around 15 min [24]) we chose not to investigate neglect symptoms other than anosognosia, as we wanted to attempt to replicate the result of a positive effect of CVS on anosognosia [24,25,26]. 

CVS was performed by irrigating the left ear with 60 cc of cold water (20°) for 30 s. During stimulation, the patient was lying on a hospital bed wearing Frenzel glasses (which prevent fixation and thereby permit the detection of stimulation-induced nystagmus), and their head was titled approximately 30° forward. In neurologically unimpaired participants, when performed successfully, this procedure produces a horizontal nystagmus with a leftward slow phase lasting about 3 min and a marked sensation of vertigo. Despite the fact that none of the RBD+ patients reported experiencing self-motion with vegetative sensations or nausea, the procedure was successful, as all patients displayed a 3-min oculomotor response after CVS. 

### 2.3. Statistical Analysis

We tested asymmetry between left and right named towns by computing an asymmetry score (left-right)/(left+right). Given the small number of participants in each group, non-parametric analyses were performed. The Kruskal Wallis test was used to compare this asymmetry score across groups, and the Dunn’s test was used for post-hoc analyses. The effect of caloric vestibular stimulation in RBD+ patients was tested using the Wilcoxon test on the number of left, right and central towns named, as well as the total number of named towns and the asymmetry scores. False-discovery-rate corrections were applied to all multiple comparisons. The effect of CVS on anosognosia is described qualitatively, as only three patients displayed anosognosia before CVS.

## 3. Results

Age did not differ significantly between the four groups (Kruskal Wallis chi square value = 4.38, df = 3, *p* = 0.22). 

### 3.1. Mental Evocation of the Map of France

Table 3 shows the median number of towns named according to geographical position plus the asymmetry score (plotted in Figure 1). A significant difference in asymmetry score was found for the four groups (χ² = 14.13, df = 3, *p* = 0.003). Post-hoc analyses showed that the asymmetry score in RBD+ patients was significantly different from each of the three control groups: RBD− (Z = 3.27, *p* = 0.002), LBD− (Z = 3.24, *p* = 0.002) and CONT (Z = 2.19, *p* = 0.029). Asymmetry scores for the three control groups were not significantly different from each other (all *p* values > 0.05). 

After CVS, the total number of towns named by RBD+ patients did not change (V value = 11.5, *p* = 0.7) but there was significantly less asymmetry (V = 0, *p* < 0.01). The reduced asymmetry was primarily due to an increase in the number of left-sided towns, as this was significantly greater after CVS (V = 2, *p* < 0.05), whereas the number of right-sided towns (V = 30, *p* = 0.11) and central towns (V = 4.5, *p* = 1) remained unchanged. Figure 2 shows individual asymmetry scores for the RBD+ group. Before CVS, six of the eight patients had large negative asymmetry scores. After CVS, these asymmetry scores were all closer to zero. Figure 3 shows the geographical locations of towns named before and after CVS by one RBD+ patient (patient 3).

### 3.2. Anosognosia

Three RBD+ patients showed signs of anosognosia before CVS. Patient 5 did not complain of any deficits but also failed to understand why no-one helped him to get up to walk. When asked where his left hand was he spontaneously held out his right hand, and only after the doctor insistently pointed to his left hand did he recognize it as his own. When asked if he could move his left hand he grabbed it with his right hand and shook it vigorously (score 3). Patient 4’s discourse before CVS was consistent with the presence of anosodiaphoria (score 2). She did not spontaneously mention anything about her arm and only after insistent questioning did she say “My arm doesn’t move well, it’s hard to make it start moving, but anyway I’m not left-handed. I help it a little bit. I always feel like I need to rub it” (score 2). Patient 3 had a score of 3 as he completely denied having any difficulties on the left side of his body (see Appendix A for the verbatim transcript of a conversation between him and the doctor before CVS). After CVS, all three patients were fully aware of their deficits (score 0). 

## 4. Discussion

The use of various mental imagery tasks where the patient is asked to describe known spaces has revealed that in some patients unilateral spatial neglect affects the representational field such that there is an inner bias and a lack of information concerning the contralesional space [3,27]. In order to further our understanding of representational neglect we asked eight right-brain damaged patients with left neglect to freely explore the map of France and name cities that they could mentally “see” before and after CVS. Unlike patients without neglect and healthy subjects, six of the eight patients with left neglect showed asymmetrical access to their mental representation of the map before CVS. This is consistent with the prevalence of representational neglect previously reported by Guariglia et al. who found that 31 out of 50 patients with some form of neglect had difficulties imagining the contralesional part of topographical places [28]. After CVS, patients spontaneously named more cities on the left side of the map and map asymmetry was significantly reduced.

We found a larger bias before CVS than when patients were given directional instructions and a fixed time to explore the map [18]. Interestingly, despite differences in the extent of the initial bias, the results after CVS were approximately the same for both versions of the map of France task, suggesting that CVS restores balance and reduces the initial bias and does not invert the imbalance when the initial asymmetry is small. Taken together with the two previous studies examining the effect of CVS on representational neglect [17,18], these results attest to the robustness of CVS’s effect, demonstrating that it does not depend upon a particular task or specific task parameters. Indeed, CVS affects mental representations that rely upon both episodic and semantic memory, under conditions of free exploration or with directional instructions. 

Consistent with the results of Rode and Perenin [18], we found that the number of named cities increased significantly on the left side of the map only. This makes it unlikely that the mechanism of action of CVS involves a change in general arousal, which would have produced a global effect, with an increase in the number of named cities regardless of their location. General arousal is also highly unlikely in light of the observation that simultaneous stimulation of the two ears does not alter visual neglect, and that ipsilesional stimulation can worsen symptoms [29].

The effect we observed after CVS could be due to an oculomotor deviation towards the neglected side [30], as it is well known that exploration of mental representations shares certain features with visual perception, such as eye movements [31]. For example, eye tracking data show that there is a correlation between gaze and city position during the map of France task [32]. While the effect of CVS reported here could be, in part, attributable to oculomotor deviation towards the neglected side, we think this unlikely as after CVS patients spontaneously began their mental exploration on the right part of their mental representation and only named left-sided cities after having first explored the right side of their representation.

Our results could be explained by CVS acting on a central, high-level mechanism. Two suggestions as to how this could occur have been proposed in the literature. First, gaze reorientation towards the neglected side might result in reorientation of attention in a way that restores the balance in attention orientation [33]. A second suggestion, drawn from the work of Ventre and colleagues, suggests that CVS corrects the deviation of the egocentric frame of reference [13,14]. Consistent with the attentional hypothesis, a previous study in healthy participants showed that vestibular stimulation induced a temporarily attentional spatial bias towards the side opposite to the stimulation [34]. However, we previously showed that patients behave similarly on the map of France task with their eyes open and closed, suggesting that the magnetic attraction of attention that occurs when the eyes are open does not affect the representational deficit, and that the deficit is representational and not attentional [35]. Furthermore, CVS reduces disorders such as anosognosia and personal neglect which do not require visual control, thus making it unlikely to act specifically on spatial attention (see review, [36]). 

Similar to previous reports [24,25,26], we showed that CVS induced a drastic remission of anosognosia in the three patients who initially exhibited unawareness of their hemiplegia and neglect. This finding reinforces the idea that CVS does not trigger neglect amelioration via a non-specific change in hemispheric activation nor by gaze reorientation, but rather by a central, high-level effect acting on the representation of self-centered space and body [17]. For example, the reduction of the directional bias after CVS might be explained by the restoration of the egocentric frame of reference. Ventre and colleagues suggested that brain injury introduces an asymmetry between bilateral associative structures causing a deviation of the egocentric frame of reference [13]. CVS might correct this deviation [14]. This idea is supported by the fact that the multisensory vestibular system is part of the network that codes for the egocentric frame of reference [37] and by the fact that representational neglect might arise from an egocentric frame of reference imbalance [11,12,38]. Indeed, patients are able to create a mental representation using an allocentric reference frame, whereas a deficit is observed when a body-centered image is formed [11,12]. In our study, we asked patients to imagine the map in front of them, forcing them to adopt an egocentric reference frame. Thus it is likely that CVS reduces asymmetrical access to mental representation by restoring the deviation of the egocentric frame of reference.

## 5. Conclusions

Together with previous studies, our results confirm that regardless of the space explored, patients with representational neglect are unaware of, and fail to explore, the contralesional half of space. Using the map of France task, we showed that this inner bias is similar when attention is directed by the experimenter and when they are free to explore the space as they choose. Indeed, even when explicitly instructed to attend to contralesional space patients cannot direct attention to the whole space in order to retrieve the corresponding semantic information. It is important to note, however, that the semantic information from this space is not lost, but becomes inaccessible because of the representational deficit. It would appear that the fact of imagining the space (where) prevents patients with representational neglect from recalling the corresponding semantic information (what), even though this information is accessible under other recall conditions without mental imagery [27,39]. 

We have shown that CVS can alleviate this inner bias, reducing asymmetrical access to mental representations both during externally-directed and free exploration of the mental space, likely through a high-level mechanism which might restore the deviation of the egocentric reference frame. It is important to acknowledge that this study did not include a placebo-controlled condition, and as such the assessors were not blind to the experimental condition. Future studies should be blinded and controlled, with the inclusion of a placebo or ipsilesional stimulation. Due to the short duration of its effect, CVS is unlikely to be exploitable for rehabilitation, but it remains a useful tool for exploring the mechanisms underlying neglect.

## Figures and Tables

**Figure 1 brainsci-10-00323-f001:**
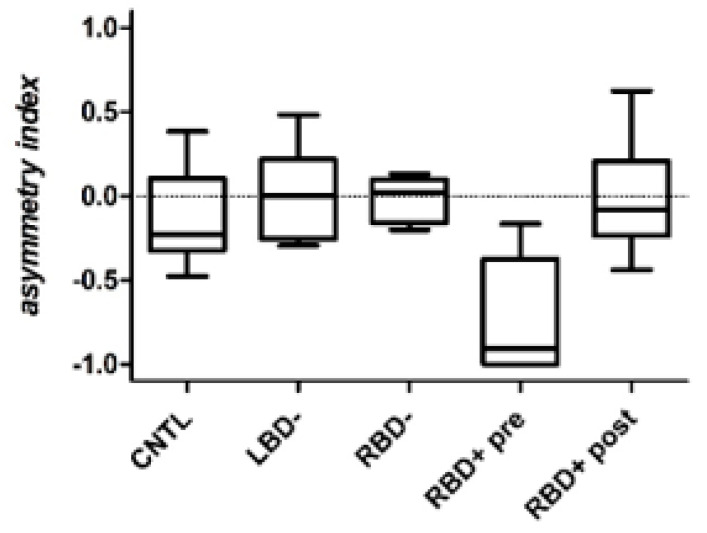
Group asymmetry index data during free exploration of the map of France for each of the three control groups and for the RBD+ group before and after caloric vestibular stimulation (CVS). Whiskers represent 1.5 X IQR.

**Figure 2 brainsci-10-00323-f002:**
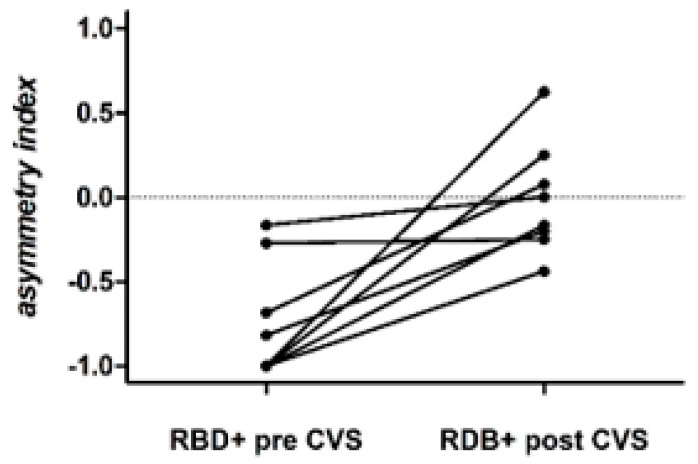
Asymmetry index data for each of the eight RBD+ patients before and after CVS.

**Figure 3 brainsci-10-00323-f003:**
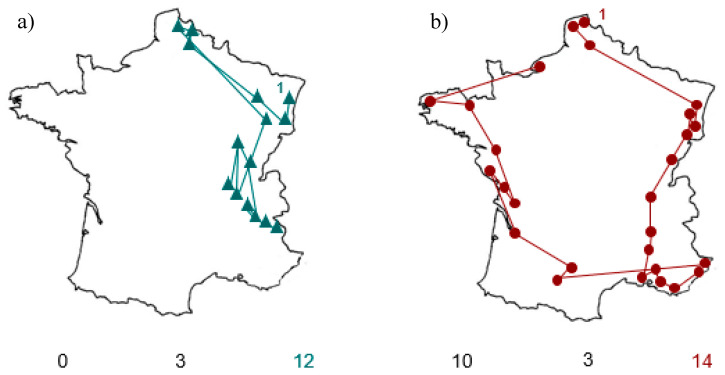
Example of towns named by patient 3 (**a**) before and (**b**) after CVS. Cities are linked following Table 1. Interestingly, those cities named on the right after stimulation were not the same as those named before stimulation, suggesting the absence of a test-retest effect. Note also, that after stimulation the patient spontaneously named cities on the right before naming left-sided cities.

**Table 1 brainsci-10-00323-t001:** Age and gender of the eight participants in each group.

	RBD+	RBD−	LBD−	CONT
**Age, median (IQR)**	64 (9.8)	65 (13.8)	64.5 (13.5)	53.5 (16.5)
**Gender**				
**Male**	6	7	6	5
**Female**	2	1	2	3

**Table 2 brainsci-10-00323-t002:** Demographic, lesion and neglect assessment data for left unilateral neglect (RBD+) patients: oculocephalic deviation (GEREN scale [22]), motor deficit assessed by clinical examination (upper-limb–lower-limb), visual field deficit assessed by visual campimetry, peripersonal neglect assessed by the Albert cancellation test [19] (number of omissions/40 (Left omissions, Middle omissions, Right omissions)) and the Schenkenberg line bisection test [20] (mean deviation in cm (number of omissions)), personal neglect (Bisiach score [21]), anosognosia (Bisiach score [23]), object centered neglect assessed by drawing a flower and a butterfly from memory, constructional apraxia assessed by drawing from memory and copying a cube. Other deficits were neglect dyslexia (reading a text), hyperschematia (previous drawing from memory) and motor perseveration on cancellation test and drawing from memory.

Case	Age/Sex	Etiology	Lesion Localization	Delay Post-Onset	Oculo-Cephalic Deviation	Left Motor Deficit	Visual Field Deficit	Albert Cancellation Test	Schenkenberg Line Bisection Test	Personal neglect	Anosognosia	Object-Centered Neglect	Constructional Apraxia	Other Deficits
1	64/M	I	Parietal, temporal, putamen	4 months	2	C-In	LHH	29 (18, 5, 7)	3.33 (8)	0	0	present	absent	-
2	51/F	I	Frontal, parietal, temporal, internal capsule, putamen	1 month	3	C-C	LHH	0	2.10 (3)	0	0	absent	present	ND
3	64/M	H	Parietal, temporal	1 month	3	C-C	LHH	36 (18, 4, 14)	-	3	3	present	present	ND
4	69/F	I	Parietal, temporal occipital	3 weeks	2	C-C	LHH	21 (18, 2, 1)	1.29 (3)	3	2	present	absent	HS
5	63/M	H	Putamen, corona radiata	5 weeks	3	C-C	-	0	4.03 (5)	3	3	absent	present	MP
6	72/M	I	Parietal, temporal	1 month	1	A-In	LIQ	8 (7,1)	−2.10 (0)	1	0	present	present	MP
7	50/M	I	Internal capsule	5 weeks	2	C-In	-	34 (18, 4, 12)	4.01 (5)	0	0	present	absent	ND
8	78/M	I	Parietal	1 month	0	In-In	LIQ	14 (9, 2, 3)	0.79 (0)	0	0	present	present	-

Abbreviations: I, ischemic; H, hemorrhagic; C, complete; In, incomplete; A, absent; LHH, left homonymous hemianopia; LIQ, left inferior quadrantanopia; MP: motor perseverations; ND: neglect dyslexia; HS: hyperschematia.

**Table 3 brainsci-10-00323-t003:** Median (IQR) number of cities named by each group according to geographical position and asymmetry score.

	LEFT	CENTRAL	RIGHT	ASYMMETRY SCORE
**RBD+ PRE CVS**	0.5 (3.5)	3.5 (1.5)	13 (6.8)	−0.91 (0.42)
**RBD+ POST CVS**	7.0 (2.5)	4.0 (2.3)	8.0 (7.3)	−0.08 (0.33)
**RBD−**	11.5 (5.8)	5.5 (3.5)	9.5 (2.8)	0.02 (0.24)
**LBD−**	10.0 (9.5)	4.0 (1.8)	12.5 (8.5)	0.0 (0.38)
**CONT**	8.0 (2.3)	4.5 (5.3)	11 (3.5)	−0.23 (0.38)

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
