# Peer review of "Caloric Vestibular Stimulation Reduces the Directional Bias in Representational Neglect"

_brainsci, 2020, doi:10.3390/brainsci10060323_

Round 1

Reviewer 1 Report

I would like to thank the Authors for taking into account my comments on the revised version of their manuscript. However, despite the correction brought by the authors, I’m still not convinced about the novelty of the results reported on the paper. As I have already stated in my first revision, this paper replicates the observation that CVS ameliorates representational neglect in a sample of a size comparable to that of the two previous studies (n=8) and using the same task used in the paper of Rode and Perenin (1994). At the theoretical level, the novelty of using a non-cued version of this task remains not clear to me. For example, did you have specific a priori hypotheses regarding the use of this version of the task?

In the discussion, on line 197-198, the Authors wrote the paper aimed “to investigate the effect of CVS on representational neglect in more patients, and using a slightly different task from that used in the two previous studies”. If that was their purpose, the sample size should be larger than that of previous studies, and the theoretical reasons for using “this slightly different task” should be explained more clearly in the Introduction and Discussion.

Moreover, the introduction remains still concise, and the hypotheses of the study are not clearly explained. Finally, I remain of the opinion that the authors should also describe the clinical/neurological (e.g. visual field deficit, line bisection test) and lesional (e.g., aetiology, lesion location) characteristics of the two control groups of RBD and LBD patients without neglect. Finally, since the Authors tested anosognosia for hemiplegia using the Bisiach questionnaire, it could be necessary also to enter in Table 2 the scores relating to the severity of the motor deficits.

Author Response

Point 1: I would like to thank the Authors for taking into account my comments on the revised version of their manuscript. However, despite the correction brought by the authors, I’m still not convinced about the novelty of the results reported on the paper. As I have already stated in my first revision, this paper replicates the observation that CVS ameliorates representational neglect in a sample of a size comparable to that of the two previous studies (n=8) and using the same task used in the paper of Rode and Perenin (1994). At the theoretical level, the novelty of using a non-cued version of this task remains not clear to me. For example, did you have specific a priori hypotheses regarding the use of this version of the task?

Response 1: We thank the reviewer for this comment. The introduction has been completely rewritten in order to better detail the theoretical context of representational neglect, effects of caloric vestibular stimulation and the working hypothesis. In particular, we now clearly state our rationale for repeating our previous experiment with the important methodological change of not giving patients any directional cues regarding their exploration of the map of France. In the previous study using the map of France the results were not the same (either before or after CVS) depending on the direction in which patients were instructed to explore the map. This suggests that directing patients' attentional orientation interferes with the representational deficit and the effect of CVS on this deficit. In the present study we wanted to remove all directional instructions in order to explore the full potential of CVS on representational neglect.

Point 2: In the discussion, on line 197-198, the Authors wrote the paper aimed “to investigate the effect of CVS on representational neglect in more patients, and using a slightly different task from that used in the two previous studies”. If that was their purpose, the sample size should be larger than that of previous studies, and the theoretical reasons for using “this slightly different task” should be explained more clearly in the Introduction and Discussion.

Response 2 : We agree with the reviewer's comment as our phrasing was ambiguous. The aim of the study was not to investigate the effect of CVS on representation neglect in more patients than in the previous studies, but to increase the total number of published cases by testing 8 right-brain damaged patients with neglect under conditions in which they were completely free to imagine and mentally explore the map of France, i.e. in the absence of directional instructions.

Point 3: Moreover, the introduction remains still concise, and the hypotheses of the study are not clearly explained. Finally, I remain of the opinion that the authors should also describe the clinical/neurological (e.g. visual field deficit, line bisection test) and lesional (e.g., aetiology, lesion location) characteristics of the two control groups of RBD and LBD patients without neglect. Finally, since the Authors tested anosognosia for hemiplegia using the Bisiach questionnaire, it could be necessary also to enter in Table 2 the scores relating to the severity of the motor deficits.

Response 3: The introduction has been completely rewritten in order to better explain the study's hypotheses and additional clinical information concerning the two patient control groups has been added in the text as has the severity of the motor deficit in the RBD+ patients in Table 2.  

Reviewer 2 Report

The authors have reported a study in which they compare four groups of 8 participants in the map of France task. Specifically, they compare RBD patients before and after CVS. The manuscript is well written. However, some minor points need to be improved.

  1. Line 27, there is a change between contralesional (previous part) and left / right (next part). A simple sentence that connects the two parts that explain the large part of SN patients are RBD could solve this point.
  2. Line 57: The authors refer to "our group." In writing a manuscript is preferable to implicit reports the authors because they are not all the same.
  3. For readers that are not expert in the field, a brief explanation of the difference between the cued and uncued map of France task could be placed in the last part of the introduction.
  4. There is a typo in the caption of table 2 "cancelation" instead of "cancellation."
  5. I appreciate the use of FDR instead of Bonferroni that drops power.
  6. About the reporting of analysis, it is preferable to report only the value of statistics and p-value, removing the test. E.g. (V=11.5, p=0.7) instead of (Wilcoxon V value =11.5, p=0.7).
  7. To improve readability, I suggest to join the two tables 3 and 4 and to build a boxplot or barplot with the asymmetry score for the four groups + post CVS for RBD patients.

Author Response

Point 1: Line 27, there is a change between contralesional (previous part) and left / right (next part). A simple sentence that connects the two parts that explain the large part of SN patients are RBD could solve this point.

Response 1: Throughout the manuscript we now explicitly state that we are referring to left neglect after right brain damage.

Point 2: Line 57: The authors refer to "our group." In writing a manuscript is preferable to implicit reports the authors because they are not all the same.

Response  2: The term "our group" has been removed.

Point 3: For readers that are not expert in the field, a brief explanation of the difference between the cued and uncued map of France task could be placed in the last part of the introduction.

Response  3.  We no longer use the term "cued" and now explain that in the previous experiment patients' exploration was timed and guided by verbal instructions from the experimenter regarding where they should begin their mental exploration. This is now detailed in the introduction.

Point 4: There is a typo in the caption of table 2 "cancelation" instead of "cancellation."

Response  4. This has been amended.

Point 5: I appreciate the use of FDR instead of Bonferroni that drops power.

Response  5. We also thought that this was the appropriate choice for these data.

Point 6: About the reporting of analysis, it is preferable to report only the value of statistics and p-value, removing the test. E.g. (V=11.5, p=0.7) instead of (Wilcoxon V value =11.5, p=0.7).

Response  6. This has been amended.

Point 7: To improve readability, I suggest to join the two tables 3 and 4 and to build a boxplot or barplot with the asymmetry score for the four groups + post CVS for RBD patients.

Response  7. This figure has been added to the manuscript.

Reviewer 3 Report

The authors investigate the effect of caloric vestibular stimulation (CVS) on representational neglect and anosognosia for hemiplegia in 8 patients with right brain damage and left-sided neglect (RBD+). First, they test the presence of representational neglect in these patients, and compare it with a group of 8 patients with right brain damage without neglect (RBD-), 8 patients with left brain damage without neglect (LBD-), and 8 healthy controls (CONT). To this aim, participants are asked to mentally evoke the map of France and name all towns they can“see” within two minutes. The number of towns at the left, center, and right side of the map are counted per participant, and an asymmetry score is computed. The authors find that there is an asymmetry in named towns in the RBD+ group compared to the other groups. Furthermore, this asymmetry reduces after one session of CVS. In addition, in three patients with anosognosia, the anosognosia completely disappears after one session of CVS. The results are striking, although I have some concerns about the methods of the study and its implications.

Title

  • The title is a bit misleading, as it answers a question that was not asked in my opinion (i.e., there is emphasis on the left hemispace, but I would think the main finding is the disappearance of the asymmetry). I doubt whether it is in line with the main result. The total number of towns, the number of central towns, and the number of right-sided towns before and after CVS did not change, however, the number of left-sided towns increased. Although the authors are correct that the change in number of left-sided towns was the only statistically significant finding, it does not logically make sense to assume that at the same time, the number of right-sided towns AND the number of total towns did not change. Looking at the data in Table 4, a change from 13 right-sided towns before CVS, to 8 right-sided towns after CVS, suggests that the total number of towns stayed the same, but the distribution left versus right was more balanced. This is a very interesting and important finding, but it does not mean that ‘CVS only affects left hemispace in representational neglect’. Also, it would be informative to emphasize that effects are only temporarily. Consider changing the title to something like: ‘CVS temporarily restores the directional bias in representational neglect’ or ‘CVS can temporarily reduce representational neglect’.

Introduction

  • In the Introduction section, the authors are describing the behaviour of left-sided neglect (since the terms ‘left’ and ‘right’ are used), although right-sided neglect can occur as well. Please change the terms left/right into contralesional/ipsilesional in the Introduction section, or explicitly mention that you are referring to left-sided neglect after right brain damage.
  • In the final paragraph of the Introduction section, the authors write that they used a ‘cued version of the map of France task’, and that they want to investigate effects of CVS on a non-cued version of the same task. As this is pointed out as the main difference between their previous study and the current study, please explain a bit clearer what is meant by ‘cued’. Also please reflect on the rotating element in the previous study, and the lack thereof in the current study.
  • In addition, please hypothesize what is expected by using this different task version, and what new insights it would provide. In other words, why is it so important to conduct this different version of the task? What will we learn from it compared to the previous studies? If the main reason for this study is replication, then please be explicit about that.

Materials and Methods

  • In what setting were these patients/controls included (which hospital?), were they admitted for inpatient rehabilitation? How were they selected; and did the authors apply any other in- or exclusion criteria?
  • Which neglect tests were used and which thresholds for neglect were used (some of the information is presented in the caption of Table 2, but please also specify this in the text of the method section)? On how many tests did patients have to show neglect in order to be included? Please include references for these tests as well.
  • The authors mention that the Albert cancellation test was used for extrapersonal neglect, why would this measure extrapersonal neglect? Please describe in more detail.
  • Was inclusion stopped when 8 people per group were obtained, or were 8 people selected from a larger sample? What was the rationale for aiming for 8 people per group (if there was any)?
  • Could the authors specify in the text that it regarded stroke patients (ischemic or haemorrhagic), as was done below Table 2.
  • Could the authors summarize stroke-related and neglect-related information for the other stroke groups as well, and add it to Table 1. Also please compare these characteristics between groups.
  • Could the authors include information about the time post-stroke patients were tested? This applies for the other stroke groups as well (that information could be presented in Table 1).
  • The information in Table 2 is useful but please consider removing the initials of the patients as this is sensitive information, especially in combination with age and sex, as it could trace back to the patients. Consider assigning numbers or letters instead of initials to refer to the patients.
  • Please explain why Frenzel glasses were worn and what they do.
  • In the statistical analysis section, please mention that you will describe scores for anosognosia before and after CVS, without doing any statistical analyses because of the small sample size (n = 3).
  • The authors only test the effect of CVS on anosognosia for the patients who show anosognosia in the first place. This makes sense. Was this also done for the mental representation? I.e., did all patients show an asymmetry in mental representation before CVS? Could the authors provide individual data of all RBD+ patients for this, and reflect on it?

Results

  • Please provide the 3 decimals of the p-value for all comparisons (unless p < .001).
  • As there are only 8 patients, it would be informative to present maps like the ones in Figure 1 for each of the patients. This could be part of Supplementary material if necessary.
  • Please also include a figure (or two figures, one for the 4 groups, and one for the pre-post measures) showing the main results (e.g. boxplots with individual data points, connected with lines to show the before and after result for the RBD+ group).
  • The data on the anosognosia is interesting but it is presented a bit messy, as it does not seem consistent to present the full interview for one patient, and only part of it for the other two. I would suggest to try to be more consistent, and present the full interviews before AND after treatment as Supplementary material, and only summarize findings in the result section. Alternatively, present data of all three patients in a similar way (e.g. one statement before, and one statement after treatment).

Discussion

  • The authors mention that the fact that they used a different task than the one used in a previous study is a novelty of the study. Please provide more information on the differences between tasks and explain why this would be important.
  • Although the results are striking, a major limitation of the study is that there was no placebo-controlled condition (or a condition with ipsilesional stimulation for example), and that presumably, the assessor of the outcome measures was not blinded (this is also not possible, as there was only one condition). Especially for the assessment of anosognosia, but also for the assessment of the cities in France, this could have influenced responses given by the patients. Please acknowledge this flaw in the study in the Discussion section, and make suggestions for future research.
  • Please also elaborate on the implications of this finding, is this providing insight in mechanisms of representational neglect or is it a potential therapy? And why would it (not) be? What would be next steps to learn more? What knowledge about the mechanisms is still lacking? Etc.
  • Please also close with a general conclusion.

Author Response

Point 1: The title is a bit misleading, as it answers a question that was not asked in my opinion (i.e., there is emphasis on the left hemispace, but I would think the main finding is the disappearance of the asymmetry). I doubt whether it is in line with the main result. The total number of towns, the number of central towns, and the number of right-sided towns before and after CVS did not change, however, the number of left-sided towns increased. Although the authors are correct that the change in number of left-sided towns was the only statistically significant finding, it does not logically make sense to assume that at the same time, the number of right-sided towns AND the number of total towns did not change. Looking at the data in Table 4, a change from 13 right-sided towns before CVS, to 8 right-sided towns after CVS, suggests that the total number of towns stayed the same, but the distribution left versus right was more balanced. This is a very interesting and important finding, but it does not mean that ‘CVS only affects left hemispace in representational neglect’. Also, it would be informative to emphasize that effects are only temporarily. Consider changing the title to something like: ‘CVS temporarily restores the directional bias in representational neglect’ or ‘CVS can temporarily reduce representational neglect’.

Response 1: The tittle has been changed according to the reviewer's suggestion, but we chose not to include the word temporarily as we did not repeat the task multiple times after CVS.

Point 2: In the Introduction section, the authors are describing the behaviour of left-sided neglect (since the terms ‘left’ and ‘right’ are used), although right-sided neglect can occur as well. Please change the terms left/right into contralesional/ipsilesional in the Introduction section, or explicitly mention that you are referring to left-sided neglect after right brain damage.

Response 2: Throughout the manuscript we now explicitly state that we are referring to left neglect after right brain damage.

Point 3: In the final paragraph of the Introduction section, the authors write that they used a ‘cued version of the map of France task’, and that they want to investigate effects of CVS on a non-cued version of the same task. As this is pointed out as the main difference between their previous study and the current study, please explain a bit clearer what is meant by ‘cued’. Also please reflect on the rotating element in the previous study, and the lack thereof in the current study.

Response 3: We no longer use the term "cued", but we were referring to the fact that in the previous experiment the patients' exploration was guided by an initial verbal instruction from the experimenter regarding where they should begin their mental exploration. This is detailed in the introduction.

In the introduction we no longer refer to the rotating element of the 1994 study as it detracted from the main ideas that we wanted to convey. However, to answer the reviewer's query, the 1994 paper was the first to use the map of France task to test representational neglect. The rotated configuration was performed to ensure that the degree of the deficit was similar in both rotations as this is an essential feature tasks that test representational neglect. Since our main interest was in the degree of the deficit during free exploration and the effect of CVS on this deficit, we chose not to include the rotation in this study.

Point 4: In addition, please hypothesize what is expected by using this different task version, and what new insights it would provide. In other words, why is it so important to conduct this different version of the task? What will we learn from it compared to the previous studies? If the main reason for this study is replication, then please be explicit about that.

Response 4. The introduction has been completely rewritten. We now clearly state our rationale for repeating our previous experiment with the important methodological change of not giving patients any directional cues regarding their exploration of the map of France. We have also included a section in the discussion comparing the results using the two versions of the tasks.

Point 5: In what setting were these patients/controls included (which hospital?), were they admitted for inpatient rehabilitation? How were they selected; and did the authors apply any other in- or exclusion criteria?

Response 5. The setting as well as detailed exclusion and inclusion criteria have been added to the methods.

Point 6: Which neglect tests were used and which thresholds for neglect were used (some of the information is presented in the caption of Table 2, but please also specify this in the text of the method section)? On how many tests did patients have to show neglect in order to be included? Please include references for these tests as well.

Response 6. The text now includes additional information, including thresholds and references, concerning the neglect tests used.

Point 7: The authors mention that the Albert cancellation test was used for extrapersonal neglect, why would this measure extrapersonal neglect? Please describe in more detail.

Response 7. The Albert cancellation test (Albert, 1973) is a validated reference test for the assessment of extrapersonal neglect, particularly the visuomotor component of the deficit (Binder et al., Verdon et al., 2010). This test also assesses body-centered neglect.

Point 8: Was inclusion stopped when 8 people per group were obtained, or were 8 people selected from a larger sample? What was the rationale for aiming for 8 people per group (if there was any)?

Response 8. We did not do an a priori calculation to determine the size of the sample. We did not exclude any patients, but rather stopped recruitment at 8 patients per group. Despite recruiting from a sizable rehabilitation service, few patients among those hospitalized in the ward were capable of performing the task. Thus, this number reflects our capacity to recruit 3 groups of brain-damaged patients whose characteristics were comparable, and who were capable of performing the mental imagery task (sufficient level of studies or general knowledge of the main cities of France, understanding of the experimental instructions, capacity for mental evocation and exploration of the map of France). It is important to note (and this information is now included in the manuscript) that RBD+ patients were included on the basis of the presence of neglect, not on the presence of representational neglect.

Point 9: Could the authors specify in the text that it regarded stroke patients (ischemic or haemorrhagic), as was done below Table 2.

Response 9. This has been added to the methods.

Point 10: Could the authors summarize stroke-related and neglect-related information for the other stroke groups as well, and add it to Table 1. Also please compare these characteristics between groups.

Response 10. Details of the two patient control groups have been added in the methods.

Point 11: Could the authors include information about the time post-stroke patients were tested? This applies for the other stroke groups as well (that information could be presented in Table 1).

Response 11. Information about the delay post-stroke in the RBD+ group has been added to Table 2 and details concerning the two patient control groups have been added in the methods.

Point 12: The information in Table 2 is useful but please consider removing the initials of the patients as this is sensitive information, especially in combination with age and sex, as it could trace back to the patients. Consider assigning numbers or letters instead of initials to refer to the patients.

Response 12. The patients' initials have been removed and replaced by numbers.

Point 13: Please explain why Frenzel glasses were worn and what they do.

Response 13. Frenzel glasses were used during the CVS as they prevent fixation and thus permit detection of stimulation-induced nystagmus. Since CVS in right-brain damaged patients does not cause dizziness or vegetative disorders, observation of the nystagmus is essential in order to verify the effectiveness of stimulation.

Point 14: In the statistical analysis section, please mention that you will describe scores for anosognosia before and after CVS, without doing any statistical analyses because of the small sample size (n = 3).

Response 14: This was added to the Statistical analysis section.

Point 15: The authors only test the effect of CVS on anosognosia for the patients who show anosognosia in the first place. This makes sense. Was this also done for the mental representation? I.e., did all patients show an asymmetry in mental representation before CVS? Could the authors provide individual data of all RBD+ patients for this, and reflect on it?

Response 15: Individual RBD+ patient data are now included in the manuscript (Figure 2). The inclusion criteria for the RBD+ group was the presence of neglect but not specifically representational neglect, which was not assessed during inclusion. This is apparent in the data from the RBD+ group, which includes two patients (4 and 5) who did not show an asymmetry on the mental representation task before CVS. This information has been added to the manuscript.

Point 16: Please provide the 3 decimals of the p-value for all comparisons (unless p < .001).

Response 16: P-value were added in results.

Point 17: As there are only 8 patients, it would be informative to present maps like the ones in Figure 1 for each of the patients. This could be part of Supplementary material if necessary.

Response 17: Individual asymmetry data before and after CVS are now included in the main text of the manuscript (Figure 2).

Point 18: Please also include a figure (or two figures, one for the 4 groups, and one for the pre-post measures) showing the main results (e.g. boxplots with individual data points, connected with lines to show the before and after result for the RBD+ group).

Response 18. The results are now displayed with a boxplot (Figure 1).

Point 19: The data on the anosognosia is interesting but it is presented a bit messy, as it does not seem consistent to present the full interview for one patient, and only part of it for the other two. I would suggest to try to be more consistent, and present the full interviews before AND after treatment as Supplementary material, and only summarize findings in the result section. Alternatively, present data of all three patients in a similar way (e.g. one statement before, and one statement after treatment).

Response 19.  We have simplified the text and the main text now includes one statement before and one after CVS for each of the three patients. The full interview from patient 3 has been included in the supplementary materials.

Point 20: The authors mention that the fact that they used a different task than the one used in a previous study is a novelty of the study. Please provide more information on the differences between tasks and explain why this would be important.

Response 20. The introduction and discussion have been substantially rewritten and now include an explanation of the importance of the current study and a comparison of the results with those of the previous study by Rode and Perenin (1994).

Point 21: Although the results are striking, a major limitation of the study is that there was no placebo-controlled condition (or a condition with ipsilesional stimulation for example), and that presumably, the assessor of the outcome measures was not blinded (this is also not possible, as there was only one condition). Especially for the assessment of anosognosia, but also for the assessment of the cities in France, this could have influenced responses given by the patients. Please acknowledge this flaw in the study in the Discussion section, and make suggestions for future research.

Response 21: These limitations are now mentioned in the conclusion.

Point 22: Please also elaborate on the implications of this finding, is this providing insight in mechanisms of representational neglect or is it a potential therapy? And why would it (not) be? What would be next steps to learn more? What knowledge about the mechanisms is still lacking? Etc.

Response 22: Perspectives and implications for this findings have been added to the conclusion.

Point 23: Please also close with a general conclusion.

Response 23: A general conclusion has been added.

Round 2

Reviewer 1 Report

The Authors addressed all the points I raised in my review letter satisfactorily, and the answers are convincing. Thus I have no further comments or points for them. In my opinion, the paper is now suitable for publication.

Author Response

We thank both reviewers for their thoughtful comments and criticisms.

Reviewer 3 Report

The study has improved substantially, and the additional information and data (e.g. on the individual patients) makes the conclusion much more convincing and stronger. The authors addressed most of my comments, and I only have few remarks left that the authors might want to consider.

  • In the Method section, it is not entirely clear what is meant by temporo-spatial disorientation, does that mean disorientation in time and space (e.g. as measured with the MMSE or MoCA questions)? Please specify.

  • The authors now describe the neglect tests that were used, this is very useful information. They also write: “Patients were considered to have neglect if neglect was detected on at least one of the tests assessing space or frame of reference.” Could the authors please add the tests they refer to (e.g. in brackets)?

  • The authors removed the explanation on the Bisach score, but since this is one of the main outcome measures it would be useful to keep this information (either in section 2.1 or section 2.2.)

  • The Albert line cancellation test is referred to as measuring extrapersonal neglect. I still think this is not the correct term to describe this test. Many papers on extrapersonal and peripersonal neglect describe it as “space outside reaching distance (far extrapersonal space), space within reaching distance (near extrapersonal or peripersonal space), and space of the body surface (personal or bodily space)” (Committeri et al., 2007), similarly described in (Aimola, Schindler, Simone, & Venneri, 2012; Halligan, Fink, Marshall, & Vallar, 2003; Van der Stoep et al., 2013). I would strongly advice to change this into peripersonal neglect. If the authors choose to call it extrapersonal neglect, please consider calling it ‘near extrapersonal neglect’ and include the distance this test was administered on to avoid any confusion.

Aimola, L., Schindler, I., Simone, A. M., & Venneri, A. (2012). Near and far space neglect: Task sensitivity and anatomical substrates. Neuropsychologia, 50(6), 1115–1123. https://doi.org/10.1016/j.neuropsychologia.2012.01.022

Committeri, G., Pitzalis, S., Galati, G., Patria, F., Pelle, G., Sabatini, U., … Pizzamiglio, L. (2007). Neural bases of personal and extrapersonal neglect in humans. Brain, 130(2), 431–441. https://doi.org/10.1093/brain/awl265

Halligan, P. W., Fink, G. R., Marshall, J. C., & Vallar, G. (2003). Spatial cognition: Evidence from visual neglect. Trends in Cognitive Sciences, 7(3), 125–133. https://doi.org/10.1016/S1364-6613(03)00032-9

Van der Stoep, N., Visser-Meily, J., Kappelle, L., de Kort, P., Huisman, K., Eijsackers, A., … Nijboer, T. (2013). Exploring near and far regions of space: distance-specific visuospatial neglect after stroke. Journal of Clinical and Experimental Neuropsychology, 35(8), 799–811. https://doi.org/10.1080/13803395.2013.824555

Author Response

We thank both reviewers for their thoughtful comments and criticisms.

Point 1: In the Method section, it is not entirely clear what is meant by temporo-spatial disorientation, does that mean disorientation in time and space (e.g. as measured with the MMSE or MoCA questions)? Please specify.

Response 1: Indeed, we meant disorientation in time and space which was assessed by questionning the patient during the clinical interview. This information has been added to section 2.1.

Point 2: The authors now describe the neglect tests that were used, this is very useful information. They also write: “Patients were considered to have neglect if neglect was detected on at least one of the tests assessing space or frame of reference.” Could the authors please add the tests they refer to (e.g. in brackets)?

Response 2: We have added this information to section 2.1.

Point 3: The authors removed the explanation on the Bisach score, but since this is one of the main outcome measures it would be useful to keep this information (either in section 2.1 or section 2.2.)

Response 3:  The Bisiach score is now described in section 2.1.

Point 4: The Albert line cancellation test is referred to as measuring extrapersonal neglect. I still think this is not the correct term to describe this test. Many papers on extrapersonal and peripersonal neglect describe it as “space outside reaching distance (far extrapersonal space), space within reaching distance (near extrapersonal or peripersonal space), and space of the body surface (personal or bodily space)” (Committeri et al., 2007), similarly described in (Aimola, Schindler, Simone, & Venneri, 2012; Halligan, Fink, Marshall, & Vallar, 2003; Van der Stoep et al., 2013). I would strongly advice to change this into peripersonal neglect. If the authors choose to call it extrapersonal neglect, please consider calling it ‘near extrapersonal neglect’ and include the distance this test was administered on to avoid any confusion.

Response 4: The Albert cancellation test was performed within reaching space and, as the reviewer correctly points out, therefore tested peripersonal neglect. This has been amended.

This manuscript is a resubmission of an earlier submission. The following is a list of the peer review reports and author responses from that submission.

Round 1

Reviewer 1 Report

Caloric vestibular stimulation only affects left hemispace in representational neglect

Hole, J., Reilly, K.T., Nash, S., & Rode, G.

MDPI Brain Sciences

GENERAL
=======

In this brief report, the authors describe the effects of caloric vestibular stimulation (CVS) on representational neglect. Specifically, they describe the reduction of an attentional bias in mental space away from the contralesional field for right-hemisphere neglect patients after CVS.

I enjoyed reading the manuscript, which is refreshingly succinct and to-the-point. The writing is clear, and the methodology and statistics are appropriate. I provided the authors address the relatively minor points outlined below.

SPECIFIC POINTS
===============

1. The complete reversal of lack of deficit awareness after CVS in all patients outlined under "3.2 Anosognosia" is quite miraculous. One could come away thinking patients' awareness of their condition is linked to whether they are currently paying attention to their contra-lesional side or not. Alternatively, one could also interpret the findings as evidence of a vestibular/attentional effect on bodily awareness.

Considering the anecdotal nature of this rather profound findings, I think it requires a bit more contextualisation. Specifically, the authors could elaborate on whether these were standardised clinical interviews, and on whether control participants had a similar lack of disease awareness (and perhaps alleviation after CVS). In addition, it would be appropriate to add a disclaimer to prevent readers over-interpreting these findings.

2. Presumably the authors had to go through an internal medical/ethical review committee's approval process. It would be good if this was explicitly referenced in the Methods section, together with approval number (if applicable). (Page 2, lines 67-69.)

3. The authors say in the Discussion "According to this hypothesis, however, we would have expected to observe a reduction in the number of cities named on the right side of the map, which was not the case." (lines 181-183) However, Table 4 shows that on average, the number of named cities on the right side of the map did reduce from 13 (SD=6.8) to 8 (SD=7.3), which is a reduction of nearly one standard deviation. While I appreciate that this difference might not have been statistically significant, it is very unlikely that the lack of a difference is statistically supported. In other words: a Bayesian test or a frequentist equivalence test is unlikely to show that the pre and post CVS number is equal.

The authors' current statement is based on a lack of a statistically significant effect, which is not the same as evidence for the opposite. Hence, the authors either need to conduct the appropriate test (aforementioned Bayesian or equivalence test), or tone down their current over-interpretation.

4. The authors' work reminded me of a paper by Dalmaijer, 2018, Vision, https://doi.org/10.3390/vision2020016 They make a case for central processing of vestibular information, and summarise some attentional work in their Discussion section. This reference, and some of the references in that paper, seem like they could aid the authors' current manuscript. I've copied the relevant references in below, of which the authors have only cited Rubens (1985). I'm sure there is more (and perhaps more relevant work), but this happened to come to mind.

- Dalmaijer, E.S. Beyond the Vestibulo-Ocular Reflex: Vestibular Input is Processed Centrally to Achieve Visual Stability. Vision 2018, 2, 16.

- Shuren, J.; Hartley, T.; Heilman, K.M. The Effects of Rotation on Spatial Attention. Neuropsychiatr. Neuropsychol. Behav. Neurol. 1998, 11, 72–75.

- Saj, A.; Honore, J.; Bernati, T.; Coello, Y.; Rousseaux, M. Subjective Visual Vertical in Pitch and Roll in Right Hemispheric Stroke. Stroke 2005, 36, 588–591.

- Kerkhoff, G.; Zoelch, C. Disorders of visuospatial orientation in the frontal plane in patients with visual neglect following right or left parietal lesions. Exp. Brain Res. 1998, 122, 108–120.

- Rubens, A.B. Caloric stimulation and unilateral visual neglect. Neurology 1985, 35, 1019–1024.

- Cappa, S.; Sterzi, R.; Vallar, G.; Bisiach, E. Remission of hemineglect and anosognosia during vestibular stimulation. Neuropsychologia 1987, 25, 775–782.

Reviewer 2 Report

In this paper, the Authors described the effect of left cold Caloric Vestibular Stimulation (LCCVS) on a group of RBD patients with neglect showing that this manipulation induces a transient remission of the mental imagery deficit associated with the left part of the imagined space. The Authors emphasized that they used a modified version of the Map test in which they do not explicitly ask the patients to name the cities on the right and the left side of the map (absence of a directional cue). 

Some comments are following.

1. The results of this study mainly confirm previous evidence concerning the specificity of LCCVS on visuospatial neglect and associated symptoms. For example, this stimulation is effective on spatial symptoms and is not on the associated aphasia in the same patient. Furthermore, evidence of LCCVS effect on right hemianaesthesia (Bottini et al., 2005) quite convincingly rule out the hypothesis of a general arousal increase mechanism. Furthermore, the absence of a directional cue does not represent a complete novelty as, in the Piazza del Duomo test (see Bisiach and Luzzatti 1978), the examiner does not ask the patient to report 'stimuli' from the left and the right side of the piazza, instead to recall 'from one side and the other' that is not a directional cue as the Authors intend in this paper. Using this task, Geminiani and Bottini (1992) have already demonstrated that LCCVS is effective in the temporary recovery from representational neglect. 

2. Concerning the discussion, some points appear to be unclear: "First, gaze reorientation towards the neglected side might result in attention reorientation in a way that restores the balance in attention orientation [14]. According to this hypothesis, however, we would have expected to observe a reduction in the number of cities named on the right side of the map, which was not the case.". Re-orientation of attention towards a previously neglected side does not necessarily imply a decrease of attention for the contralateral side (it is a balance restoration and not an inversion of the imbalance from the left side to the right side). 

3. Finally, I cannot understand the theoretical relationship the Authors propose between the transient recovery of anosognosia and the transient remission of the selective representational deficit. They refer to the theory of Ventre that explains neglect as a consequence of the disruption of the egocentric frame of reference. While this theory may explain the remission of anosognosia and symptoms associated with a deterioration of the body representation in the brain, I do not really understand how the representation of the environment can depend on the egocentric frame of reference, especially considering that the Map test does not imply any imagined motor activity of the patient within the environment.

4. Why the Authors tested an LBD group as a control? Did they correct the multiple comparisons using a Bonferroni approach?

5. Between-groups differences in asymmetry scores may be driven by a possible difference in years of education. Did the Authors control for this?

Reviewer 3 Report

In this paper, the Authors provide evidence in favour of the effectiveness of Caloric Vestibular Stimulation (CVS) in reducing representation neglect using a non-cued version of the map of France task, i.e., without giving participants any cues related to the left or right side of the map. They found that CVS ameliorates representational neglect reducing the right-left asymmetry observed before the stimulation.

My main concern is about the novelty of the results reported in the present paper. Indeed, this study replicates the already well-known observation that left cold CVS can ameliorate representation neglect. The main difference between this and previous studies is that here the experimenter does not provide participants with any directional cues related to the left or right side of the mental map. It is not clear to me what is the theoretical advantage and relevance resulting from using this kind of non-cued procedure in comparison with the classic version of the task. Therefore, in my knowledge in the “Piazza del Duomo” task for example patients are asked to describe the square according to a precise viewpoint (i.e., to imagine themselves standing in front of the cathedral or from the opposite prospective) but the experimenter does not give any cues related to the left or right side of the place. Based on these considerations, the Authors should explicitly clarify which data are a confirmation of previous studies, and what the findings of the present experiment add to those previously reported in the literature.

Moreover, in the discussion (page 6, line 162), the Authors wrote: “our results complement and strengthen previous results”. Can the Authors better explain in what way their results “strengthen” previous observations on the effect of CVS on representational neglect?

Finally, also the observation that CVS can ameliorate anosognosia for motor deficits is not new and has been already confirmed in different previous studies. Since the study aims to investigate “the effect of CVS on representational neglect”, can the Authors specify why they also included the assessments of anosognosia before and after CVS, and they did not asses the effect of CVS on other symptoms such as personal neglect?

Several other points should be carefully addressed:

Introduction

The introduction is very concise. Some concepts need to be clarified and treated in more detail. For example, on page 1 (line 32) the Authors wrote: “This suggests a deficit in generating or exploring the contralesional side of imaginary representations and an internal ipsilesional bias in spatial attention orientation”. The Authors should better explain the differences between these different explanations of representational neglect (i.e., a deficit in generating versus a deficit in exploring the internal mental representation) and their theoretical implications.

Also, a more detailed description of the different hypothesis about the effectiveness of CVS on representational neglect could be useful for readers who are not familiar with this topic.

On page 1 (line 34-42), the Authors cited two studies that investigated the effect of CVS on representation neglect but only the study of Rode and Perenin (1994) is described in detail. The Authors should describe in more detail also the study of Geminiani and Bottini (1992) in particular respect to the procedure used to test representation neglect.

Materials and methods

In Table 1, please indicate which test was used for the evaluation of visual field deficits, drawing for memory, constructional apraxia, neglect dyslexia and hyper-schematia. Please, also specify how motor perseveration was assessed.

In Table 1, the Authors reported in a single column the number of omissions in the Albert cancellation test and the % of deviation in line bisection. For clarity, I think that the results of the two tests should be reported in separate columns. Moreover, to give the reader more detailed information about the severity of the neglect, I believe it would be useful to report for the Albert test the number of omissions on the right and on the left, or an index of severity of neglect, i.e., the Center of Cancellation (CoC index; Rorden and Karnath (2010). A simple measure of neglect severity. Neuropsychologia. 48(9):2758-63).

The Authors should also include in Table 1 the clinical and neuropsychological data of patients belonging to the other two groups (RBD- and LBD-).

On page 2 (line 67), the Authors specify that patients “participated in the experiment in the 6 months following their brain injury […]”. I think that in Table 1, it could be useful to indicate for each patient the time between the onset and the neuropsychological testing.

On line 29, the Authors wrote: “The main finding of this study was that CVS resulted in a temporary remission of representational neglect”. If I understood correctly, the evaluation of representational neglect has been carried out twice: once before and once after the vestibular stimulation. How can the Authors say that the improvement is temporary? Do the Authors have data about the length of the CVS effect on representational negelct?

Minor points:  

Were the groups comparable for the participants’ education?

Please, indicate whether the task for the evaluation of representational neglect was performed with eyes closed or open.